# Stiffness Enhancement, Anti-Aging, and Self-Forming Holes in Polycarbonate/Acrylonitrile-Styrene-Acrylic by the Core-Shell Structure of Acrylic Resin

**DOI:** 10.3390/polym14040782

**Published:** 2022-02-17

**Authors:** Ji Huang, Chunliang Kuo, Hung-Yin Tsai

**Affiliations:** 1Department of Power Mechanical Engineering, National Tsing Hua University, 101, Section 2, Kuang-Fu Road, Hsinchu 300044, Taiwan; s107033871@m107.nthu.edu.tw; 2Department of Mechanical Engineering, National Taiwan University of Science and Technology, No.43, Keelung Road, Section 4, Taipei 106335, Taiwan; chunliang.kuo@mail.ntust.edu.tw

**Keywords:** self-forming hole, mechanical properties, microstructural analysis, anti-aging, impact wrinkles

## Abstract

Currently, polycarbonate/acrylonitrile-styrene-acrylic (PC/ASA) is used mainly in the automotive, outdoor electronic equipment, sports equipment, and medical care fields, but its use is limited by its poor impact resistance and aging characteristics. This study investigates the preparation of polycarbonate/acrylonitrile-styrene-acrylic/acrylic resin (PC/ASA/ACE) via melt blending. We observed that the addition of acrylic resin (ACE) enhanced the impact strength (up to 14.1%) and abrasion resistance (up to 35.7%) of the blends compared to PC/ASA. The microstructure of the copolymer was observed by scanning electron microscopy and laser scanning confocal microscopy. They were found to have a self-forming hole phenomenon, which is more favorable for potential PC/ASA applications. Furthermore, ACE addition effectively enhances the copolymer toughness and wear properties but slightly reduces their hardness, tensile strength, and melt flow rate, improving their suitability for use in applications such as aircraft windshields. After 80 cycles of aging, the PC/ASA/ACE also outperformed the impact strength of the unaged PC/ASA. ACE addition to PC/ASA can create materials with better impact and aging resistance.

## 1. Introduction

Acrylonitrile-butadiene-styrene (ABS) is one of the most widely used thermoplastic engineering plastics, with excellent comprehensive performance. However, the butadiene in ABS contains unsaturated C=C double bonds. Furthermore, the double bonds tend to break easily, causing the mechanical properties to deteriorate and change color. To address the problem of poor weather resistance, researchers replaced butadiene with poly(butyl acrylate) (PBA) in ABS and developed acrylonitrile-styrene-acrylic (ASA) in the 1970s [1]. There are no double bonds in ASA, so it is superior to ABS in terms of its weather and chemical resistance [2]. Polycarbonate (PC) is a thermoplastic engineering plastic with excellent comprehensive performance. It has been widely used in electronics and automobile manufacturing [3,4,5,6]. PC/ASA combines the excellent properties of each component, enhancing heat resistance, impact strength, and tensile strength, and improving the processability and weather and solvent resistance of PC [7,8].

PC/ASA is increasingly used in aircraft structural parts, such as windshields and cabin covers, as an excellent outdoor material. Considering the harsh conditions that it may face, such as high wind loads, UV aging resistance, and the ability to resist bird impacts, it is vital to improve the impact toughness, wear resistance, and service life of PC/ASA by adding additives.

Elastomers have been used for many years for toughening of materials. The primary mechanism of elastomer toughness is its low modulus of elasticity, which initiates the formation of cracks and the absorption of large amounts of energy from external loads [9]. Hwang and Kim [10] reported that the addition of chlorinated polyethylene (CPE) could drastically improve the impact strength of styrene-acrylonitrile (SAN), which is a component of ASA. Powder nitrile butadiene rubber (PNBR) and hydrogenated acrylonitrile-butadiene rubber (HNBR) are elastomers that have been shown to enhance the toughness of ASA/SAN [11]. ACE is a typical impact modifier with a core-shell structure. It is particularly effective in improving the impact performance of polyvinyl chloride (PVC) [12,13,14]. Therefore, incorporating ACE into PC/ASA can be expected to improve the impact toughness. To the best of our knowledge, no study about PC/ASA/ACE has been reported.

In this study, ACE is used to enhance PC/ASA (70/30), and the mechanical properties of the blend, such as tensile and impact strength, are studied. Dynamic mechanical analysis (DMA) and Fourier transform infrared spectroscopy (FTIR) are used to measure the glass transition temperature (T_g_) and the interactions between components of the blend. With the aid of scanning electron microscopy (SEM) and laser scanning confocal microscopy (LSCM), the morphology and roughness of the PC/ASA impact sections are also investigated. At the same time, aging tests are conducted to study the service life of the material under UV irradiation. We also extend the corresponding applications to the discovered self-forming hole phenomena.

## 2. Materials and Methods

### 2.1. Materials

Extrusion grade ASA (PW-957) was provided by the ChiMei Corporation, tainan, Taiwan. When the experiment was conducted at 220 °C with a load of 10 kg, the melt volume rate was 26 cm^3^/10 min. Extrusion grade PC (PC-110) was provided by the ChiMei Corporation, tainan, Taiwan. When the experiment was conducted at 300 °C with a load of 1.2 kg, the melt volume rate was 10 cm^3^/10 min. The modifier, ACE (M-190), was provided by the Kaneka Corporation, Tokyo, Japan.

### 2.2. Sample Preparation

When ASA was dried at 70 °C for 8 h, its moisture content was <2‰. When PC was dried at 80 °C for 8 h, its moisture content was <2‰. Dried raw material particles and ACE were uniformly and proportionally mixed, and the experimental formulation is shown in Table 1. The resulting materials were put into a twin-screw extruder to melt, blend, extrude, and granulate, with extrusion temperatures ranging from 245 to 260 °C, as shown in Table 2. To prepare ISO standard splines, uniformly granulated PC/ASA/ACE particles were first dried at 80 °C for 8 h and then loaded in an injection molding machine with the temperature settings shown in Table 3. Standard splines were processed into rectangular splines (63.5 × 12.7 × 3.2 mm) to perform impact and DMA tests. To perform tensile tests, standard splines were processed into dumbbell-shaped splines.

Accelerated aging tests were performed in a UV/condensation aging chamber equipped with a UV-340 fluorescent lamp. Samples were run with UV irradiation in water condensing conditions for 8 h UV at 60 °C (UV) and 4 h in the dark in condensing conditions at 50 °C (CON). One cycle thus consisted of 8 h UV and 4 h CON.

### 2.3. Material Measurement

DMA refers to the use of a dynamic mechanical analyzer (DMA Q800, Taiwan TA Instrument Inc., Taibei, Taiwan) in torsional mode between −60 and 160 °C, with a heating rate of 3 °C/min and a frequency of 1 Hz. A melt flow index tester (LMI 4000, Dynisco, Franklin, MA, USA) was used to measure the melt flow rate (MFR) of PC/ASA/ACE. Following standard ASTM D 1238, the temperature for the test was set at 220 °C, and the load used was 10 kg [15]. A thermal analyzer (2-HT, Mettler-Toledo, Greifensee, Switzerland) was used to evaluate the thermal stability of the modified PC/ASA with the accuracy of ±1 K. The quantity of each set of samples used was 8–10 mg, the atmosphere selected was N_2_, the test temperature was set from 25 to 600 °C, and the heating rate was set as 10 °C/min.

This experiment evaluated tensile strength, impact strength, and toughness. The impact strength test was conducted using an impact tester (GT-7045-MDH, Gotech Testing Machines Inc., Taichung, Taiwan) at 25 °C and according to the standard ISO 180. The instrument has an impact angle of 150° and an energy loss of less than 2.5%. A universal testing machine was used to test the tensile strength according to the standard ISO 527, and the stretching rate was 1 mm/min. For the tensile experiment, the limitation of the utilized load cell is 1 kN and the accuracy is ±1 N. In addition, Shore D hardness was measured using a hardness testing machine (MV-1, Matsuzawa Co., Ltd., Akita, Japan) according to ISO 868:2003.

The interactions between the components of PC/ASA/ACE were studied using FTIR. An FTIR spectrometer (Vertex 80v, Bruker, Berlin, Germany) was used with a 4 cm^−1^ resolution, with each spectrum being scanned 32 times, and the test range was from 600 to 3500 cm^−1^. The SEM was used to observe the blend microstructure (JSM-7000F, JEOL, Akishima, Japan). Before the tests, a sputtering machine was used to coat a thin conductive layer of gold on the surface of the impact-cracked material. The roughness was characterized using LSCM to obtain a three-dimensional (3D) surface map. The 3D images were analyzed using VK-H2X series multi-functional analysis software. Finally, an abrasion tester was used to conduct abrasion tests. The initial weight of the sample was recorded as W_1_, and the weight at the end of the test was W_2_. The number of rev/min was 72, and the number of revolutions was 8000. The test temperature was 24 ± 2 °C, and the weight loss ratio was defined as the difference between W_1_ and W_2_ divided by W_1_.

## 3. Results

### 3.1. Material Compatibility

It is generally believed that DMA shows the glass transition more clearly than differential scanning calorimetry. DMA is used to test the T_g_ and its storage modulus. The storage modulus is the modulus of the elastic part of the material, while the loss modulus is the modulus of the viscous part of the material. Tan *δ* represents the comparison between the loss modulus and the storage modulus, while tan *δ* peaks (usually reported as T_g_) best indicate the motion of the macromolecular chain. As shown in Figure 1b, the storage modulus of the blends significantly decreased at high temperatures compared to the pure material [16]. As shown in Figure 1c, the loss modulus’ peak maximum corresponds to the onset of significant segmental motion of the polymer chain. In particular, the higher the content of ACE in all samples, the faster the decrease of the storage modulus. Figure 1a shows the tan δ curve of PC/ASA/ACE, while the inset in the top left corner of Figure 1a is an enlarged diagram of the material behavior observed through DMA when the test temperatures range from −50 to −20 °C. The results show that there is one T_g_ in the low-temperature range and two in the high-temperature range. Moreover, the level of the tan *δ* peak also represents the T_g_ of the materials. The higher tan δ peaks represent the improved energy-dissipating ability of the materials [17].

In a blended system, if the components are immiscible, the T_g_ value of each polymer will not change. However, if the components are miscible, there will be a single typical T_g_ for the blend. In general, ACE as a core-shell polymer has two T_g_ peaks. At a low temperature, T_g1_ represents the peak temperature of the rubber phase of PBA. At the same time, T_g2_ is the common SAN contained in ASA and core-shell materials, and T_g3_ represents the peak temperature of PC at high temperatures. From Table 4, we found that after adding ACE to PC/ASA, the T_g_ values of the three glass-transition temperatures (T_g_) had a similar change. The addition of ACE increased the T_g_ of ASA while decreasing the T_g_ of the PC, causing ΔT_g_ to fall [18]. After adding 5 wt.% ACE, there was a significant decrease in T_g3_, but with the increase of ACE content, there was no noticeable change in T_g3_. It indicates that PC has some compatibility with ACE; however, the compatibility is low. Five wt.% ACE saturated the solubility, so increasing the ACE content did not significantly affect the T_g_ of PC. Moreover, Table 4 shows that the change in ΔT_g12_ was greater than in ΔT_g23_. This is because both ACE and ASA contain SAN, and based on the dissolution principle in a similar material structure, the two materials are more compatible. Therefore, T_g2_ and T_g1_ are more willing to move closer to each other.

### 3.2. Change of the Melt Flow Rate

The MFR refers to the weight of the thermoplastic melt that passes through a specified standard caliber every 10 min at a specific temperature and load (measured in g/10 min). A material with a high MFR value has a high flowability. At 150 °C, DMA showed that the viscoelastic properties were already equal, and theoretically, there should not be much change in flowability at 220 °C [19]. The experimental results are shown in Figure 1b and Figure 2. The MFR significantly declined as the content of ACE increased. In particular, the decrease was most pronounced when ACE was added at 10 wt.%. After the addition of ACE, the flows of the three materials interfered with one another, resulting in a decrease in the overall flow properties of the blend. On the other hand, the PBA in ACE is rubber, and its viscosity will substantially affect the flow of the material. After adding 10 wt.% ACE, the MFR did not change significantly, which may be because the MFR of 5.5 g/10 min already reached the viscosity of ACE, and further addition could no longer affect the viscosity of PC/ASA/ACE. The area under the tan δ curve in Figure 1a is related to the magnitude of the activation enthalpy for the relaxation of the motion of the macromolecular chain. The addition of ACE increased the peak value of tan δ, indicating a high degree of energy dissipation. It demonstrates that the movement of macromolecular chains is more restricted and responsible for the fluid’s lower apparent viscosity.

### 3.3. Thermal Stability

The TG diagram can be used to compare the effective thermal stabilities of different polymers. The data from the TG diagram can provide a theoretical basis for selecting the processing temperature of the blend material. The initial degradation temperature of PC/ASA was 386.8 °C, and that of ACE was 315.5 °C, both of which were higher than the processing temperature of the material. Thus, when processing PC/ASA melt, the degradation of the blend is negligible. The initial blend comprises 30 wt.% ASA and 70 wt.% PC, and then ACE is added to the blend in different proportions. Figure 3 shows the respective TG curves of the PC/ASA/ACE. The weight loss ratio refers to the weight of a substance before drying divided by the weight of the same substance after drying. The thermal stability of the whole material changes after the addition of ACE. From Table 5, it can be seen that as the content of ACE increased in PC/ACA/ACE, the decomposition temperature of the blend gradually decreased. The thermal stability was measured using the thermal weight loss temperature (TG 5%), as shown in the enlarged image in Figure 3, where the red line represents the temperature at a weight loss of 5%. After the addition of ACE, the initial degradation temperature of decomposition decreased, but not significantly, from 386.8 to 379.2 °C. In the range from 400 to 500 °C, the degradation temperature of the material moved slightly toward the low-temperature region with the addition of ACE. In addition, in this temperature range, the composite slope gradually increased with the increase of ACE content, tending progressively to the slope of ACE. The increase in slope represents a decrease in the material’s thermal stability, indicating that the material is more susceptible to decomposition by thermal effects.

### 3.4. Mechanical Properties

Figure 4a shows the relationship between the tensile strength, impact strength, and hardness of PC/ASA and the amount of ACE added. Figure 4b shows the results of stress–strain curves before and after doping with ACE. Toughness is the ability of a material to absorb energy in plastic deformation before fracture. The integral area under the stress–strain curve can effectively represent the toughness of a material. The gray columns in Figure 4a show the tensile strength of PC/ASA/ACE at different weight ratios. The ultimate tensile strength of PC/ASA/ACE was reduced by 22.5% after adding ACE. When the content of ACE is 20 phr, the decline in the tensile strength is not as significant as for lower phr values. The decrease in the tensile strength is mainly caused by the low strength and modulus of ACE, such that when large amounts of ACE were added, the tensile strength significantly declined. However, the elongation of PC/ASA/ACE increased with the addition of ACE. Since ACE is a malleable material with good toughness, it developed a large deformation during the stretching process without rupture. Consequently, the elongation at the break of the material was increased to obtain tremendous strain energy, which improved the toughness. The material toughness increased from 58.7 to 81.9 J/m^3^, as shown in Table 6. The increase in toughness also mirrors the rise in impact strength to some extent.

The yellow columns in Figure 4a show the impact strength of the PC/ASA/ACE at different weight ratios. ACE addition effectively enhanced the toughness of the PC/ASA (70/30). As the ACE content increased, the toughness increased, and the impact strength of the blended material increased as well. In this study, the addition of ACE increased the impact strength by only 14.1%, with the highest value being 54 kJ/m^2^ at 20 wt.% ACE, because the PC/ASA (70/30) has good impact strength. Previous studies have shown that rubber phase PBA exists in ASA in stress concentration points. ACE addition brings in more rubber phase to the PBA, which serves as the point of stress concentration, induces cracking, and absorbs shock energy [20,21]. As shown in Figure 5, ACE particles in the blend can act as stress concentration points, leading to the formation of many yield zones and holes when impacted, thus consuming a large amount of impact energy and improving the impact strength of the material. However, because the ACE structure does not match the modulus of PC and ASA, the effect of poor stress transfer occurs after absorbing kinetic energy. The limited effect of stress transfer makes a hole-like structure appear after the impact, and the self-forming hole appears (as shown in Figure 5). These notches can be applied to the stamping and bending die, and the screw can be locked without drilling and positioning steps compared to conventional materials. When 30 phr of ACE was added, the impact property of the blend was weaker than that of the 20 phr ACE. Due to the core-shell structure of ACE, excessive stress concentration will affect the overall toughening effect of the material.

The blue columns in Figure 4 show the hardness of the PC/ASA/ACE at different weight ratios, and the addition of ACE slightly reduced the overall hardness of the material. This is because ACE, as an elastomer, has a low modulus and undergoes large deformation when subjected to pressure, resulting in a reduction in hardness. The toughness of the material is related to the impact properties and not to the tensile properties and hardness. The results showed that the hardness of PC/ASA/ACE (70/30/0–30) decreased by 17.9% relative to PC/ASA.

### 3.5. Abrasion Properties of Materials

Figure 6 shows the experimental results of abrasion testing when different amounts of ACE were added. The results indicate that although PC/ASA offers good wear resistance, the addition of ACE significantly improved the wear resistance of the blend. Without the addition of ACE, the weight loss of the PC/ASA was 7.2 wt.%. The weight loss of the PC/ASA/ACE was 6.5 wt.% at a 10 wt.% content of ACE. ACE is a type of core-shell material whose core contains highly elastic PBA rubber particles. Its doping improves the abrasion resistance of the material. It is a characteristic of the elastomer that when subjected to external compression, it undergoes a small deformation without rupture, thus achieving the purpose of resistance to abrasion, as shown in Figure 5. However, the weight loss of the PC/ASA/ACE increased as the content of ACE continued to increase. This suggests that the increase in the ACE content does not necessarily enhance the abrasion properties of the blend.

### 3.6. Interactions between Materials

FTIR analysis was used to understand the interaction between PC, ASA, and ACE. If there is a chemical reaction between the polymers, new chemical bonds will be formed. If not, no new chemical bonds are formed, which means that there is no intermolecular reaction between the three substances, but only similar compatibility of the materials. Figure 7 shows different weight ratios of the FTIR spectra within the range from 600 to 3500 cm^−1^.

The characteristic peaks under various ACE weighting are shown in Figure 7. Each peak is attributed as follows: 2966 cm^−1^ represents Methyl-CH_3_, 2850 and 2925 cm^−1^ represent -CH_2_ [22], 2237 cm^−1^ represents C≡N of acrylonitrile in ASA [23], and 1774 and 1200 cm^−1^ are the unique carbonate keys of PC materials. The peak at 1735 cm^−1^ is related to the stretching vibration of carbonyl (C=O) [24,25,26], the peak at 1602 cm^−1^ is the characteristic peak of benzene skeleton ring vibration [27], and the peak at 1453 cm^−1^ is the asymmetric stretching and bending vibration of >CH_2_. The characteristic peaks of disubstituted benzene are 888, 831, and 764 cm^−1^. The characteristic peak of the monosubstituted benzene ring is at 701 cm^−1^ [28]. The combination of FTIR spectra and the characteristics described above revealed no significant change in the wavenumber of the main group. This indicates no significant chemical interaction between the molecules of PC/ASA/ACE and the addition of ACE results in purely physical changes.

### 3.7. Scanning Electron Microscopy Analysis and Roughness Testing

The microstructure of the blend was observed via SEM and LSCM. The impact fracture morphology under SEM is shown on the left side of Figure 8, and the impact fracture morphology under LSCM is shown on the right side. The significant differences in the fracture’s impact surface can be observed by comparing SEM images from Figure 8, which suggests that the toughness of the copolymer, PC/ASA (70/30), can be significantly enhanced with ACE doping. The significant differences on the impact surface of the fracture can be observed through the comparison of SEM images from Figure 8, which suggests that the toughness of the copolymer, PC/ASA (70/30), can be greatly enhanced with ACE doping. The SEM image of the PC/ASA/ACE (70/30/0), in Figure 8a, shows a flat, clear surface [29]. After the addition of ACE, the surface of the blend became rough, and the transformation of the fracture surface continued with the increase of ACE content. The rise of ACE content made impact wrinkles gradually appear in the material, and the yellow line direction in Figure 8 is the cracking path. The fracture spacing between impact wrinkles gradually decreased with the increasing ACE content. The appearance and intensification of the impact wrinkles is the main reason for the roughness of the impact fracture surface. To prove the relationship between the roughness of the material section and the impact performance, LSCM was used to characterize the roughness of the material section with different ACE contents. The concrete parameters of the roughness of these materials are listed in Table 7. The parameters include arithmetic mean height (S_a_), the maximum height of profile (S_z_), and the developed area ratio (S_dr_) [30]. These parameters define the roughness of the different surfaces according to ISO 25178. The numerical changes of S_a_, S_z_, and S_dr_ are consistent with the changes in impact performance measured by SEM—they all increased with the ACE content. The increase in the roughness of the section of the LSCM image with the impact strength, resulting in more impact wrinkles, was confirmed. The SEM and LSCM analyses established the relationship between the microstructure and the properties, and the results of the analysis are consistent with the mechanical properties of the blend.

The main image shown in Figure 9 is an SEM image of a cross-section of the impacted section, viewed at 10,000 times magnification. The left view of the impact fracture in the upper right corner observes the port depth. The bottom left is a magnified SEM image of the center of the fracture. The impact bearing surface of the PC/ASA without ACE doping is flat, and the formed holes (red label) are small and sparse, indicating that the material is brittle fracture. With the increasing ACE doping, the impact surface became progressively rougher, and a notch appeared in the center: the formed holes were denser, and the peeling area (blue label) gradually increased, indicating that the fracture characteristics of the material are biased toward tough fracture. With the increasing ACE doping, the impact-bearing surface section depression deepened, forming an impact notch, and a self-forming hole phenomenon appeared.

### 3.8. The Effect of Accelerated Weathering Test

Figure 10a shows the impact strength variation of the hybrid composites under accelerated aging conditions. It was established in Section 3.4 that the composite with 20 wt.% ACE had the highest impact strength of 54 kJ/m^2^. Therefore, 20 wt.% ACE is expected to maintain the highest impact strength after aging. After 80 cycles of accelerated aging, the impact strength of all the blends was reduced. This occurred because the molecular chains of the material were broken during the aging process. PC is affected by UV light, which generates fine holes containing CO_2_ inside the material, creating areas of stress concentration that reduce the impact performance of the material. In Figure 10b, after 80 cycles of aging, the PC/ASA had the steepest slope, which indicates that it had the fastest rate of material degradation. Additionally, Figure 10b shows that the impact strength of PC/ASA decreased by 19.4%. The impact properties of the materials doped with 5 wt.% ACE, 10 wt.% ACE, 20 wt.% ACE, and 30 wt.% ACE decreased by 9.1%, 10.2%, 11.6%, and 10.5%, respectively, all of which were less than those of PC/ASA. This is because ACE does not contain ester groups that are easily cleaved by UV light. The addition of ACE reduced the PC content of the compound material, thus reducing the breakage of molecular chains. As the material ages, the thermal stability of the PC/ASA/ACE changes. In the range of 350–500 °C, the material’s thermal degradation temperature (Figure 10c) moved towards the low-temperature region as the material aged, and the temperature change of each component was not significant. Carboxyl groups were formed during PC photodegradation, and the carboxyl index (relative content of carboxyl groups) is a valuable parameter to quantify the degradation of this polymer. The carboxyl index, the ratio of carboxyl end-group absorption (peak at 3290 cm^−1^) to the reference peak (center at 2970 cm^−1^), was measured from FTIR data (Figure 10d), and the results are shown in Table 8. PC/ASA/ACE has a lower concentration of increased carboxyl index compared to PC/ASA. The main reason for the changes in slope, thermal stability, and carboxyl index is that the macromolecular chains of PC/ASA/ACE, exposed to UV radiation, undergo a lot of chain breakage. After 80 cycles of aging, the PC/ASA/ACE also had impact strengths higher than that of the unaged PC/ASA. These results indicate that hybrid composites’ interfacing improves the aging resistance of PC/ASA/ACE to UV and prolongs the service life of the material.

## 4. Conclusions

According to the results of the DMA, there are three glass transition temperatures in the different weight ratios of PC/ASA/ACE. One of them is at a low temperature, two are in the high-temperature region, and the values of the glass transition temperatures do not significantly vary. TG analysis showed that the addition of ACE had little influence on the thermal properties of the blends but reduced the degree of crystallization. The FTIR analysis revealed no chemical interaction between PC/ASA/ACE but more of a physical change. Additionally, the SEM and LSCM analysis of the impact fracture morphology revealed no apparent separation in the copolymer. The impact fracture of the material was gradually roughened with the improved impact performance. Furthermore, a pore structure was found in the center of the impact cross-section, called the self-forming hole phenomenon. The appearance and density of impact wrinkles were observed as the main reasons for the roughness of the impacted fracture.

The addition of ACE effectively enhanced the PC/ASA toughness, weather resistance, and wear properties, but slightly reduced their hardness, tensile strength, and MFR. When 20 phr of ACE was added, the toughness and wear of the copolymer significantly increased, by up to 14.1% and 35.7%, respectively, while other properties slightly declined. The PC/ASA/ACE impact performance aged by 80 cycles was stronger than that of the unaged PC/ASA. Self-forming holes can be applied to bending and stamping dies, reducing the number of processing steps, and the ability to lock screws without the need to drill holes for positioning, broadening the application of this material.

## Figures and Tables

**Figure 1 polymers-14-00782-f001:**
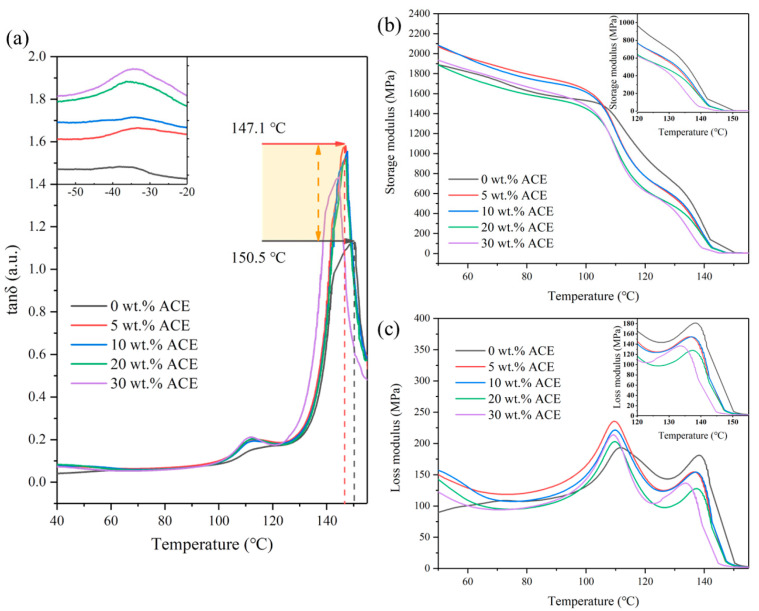
DMA curves of PC/ASA/ACE: (**a**) tan *δ*, (**b**) storage modulus, and (**c**) loss modulus.

**Figure 2 polymers-14-00782-f002:**
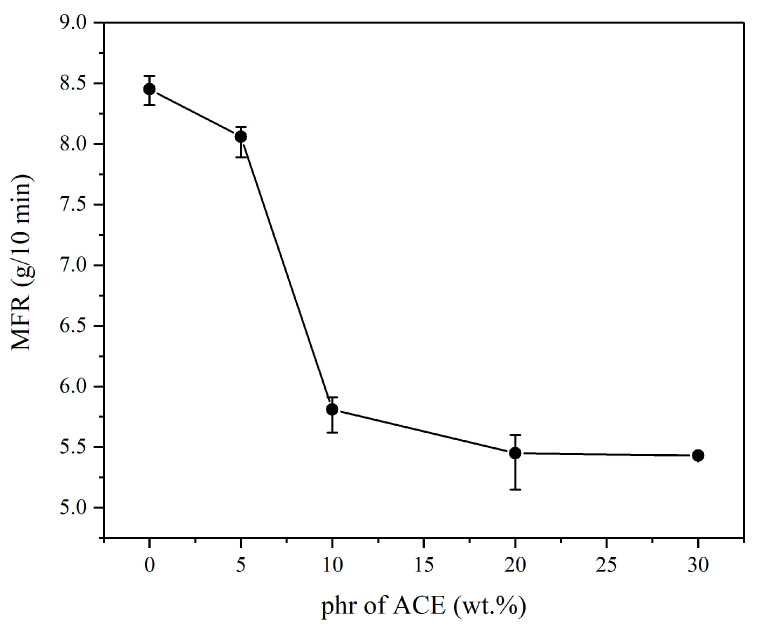
MFR of PC/ASA/ACE.

**Figure 3 polymers-14-00782-f003:**
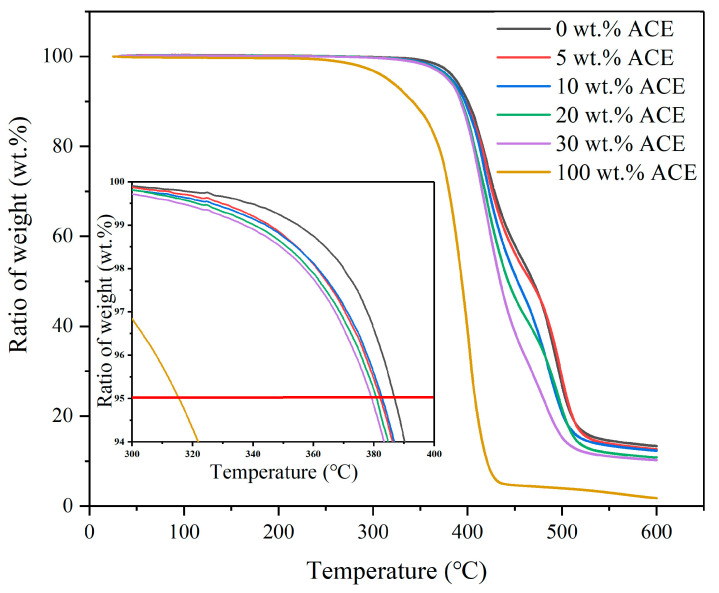
TG curves of PC/ASA/ACE.

**Figure 4 polymers-14-00782-f004:**
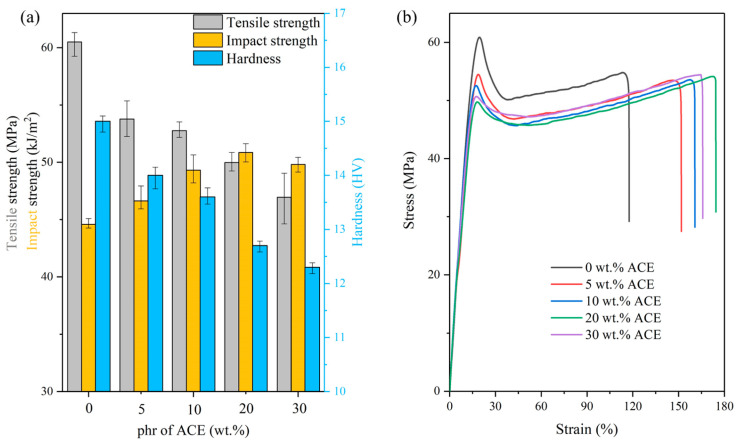
PC/ASA/ACE: (**a**) tensile strength, impact strength, and hardness, and (**b**) stress–strain curve.

**Figure 5 polymers-14-00782-f005:**
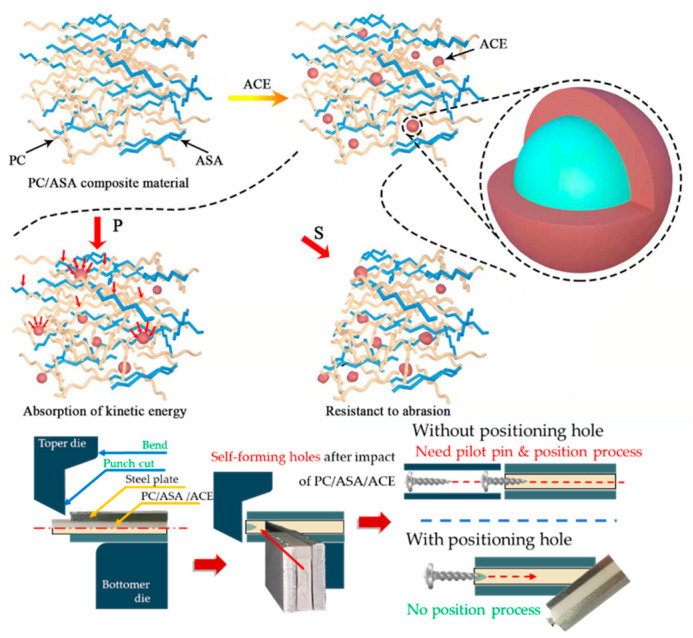
Schematic of the enhancement mechanism of ACE.

**Figure 6 polymers-14-00782-f006:**
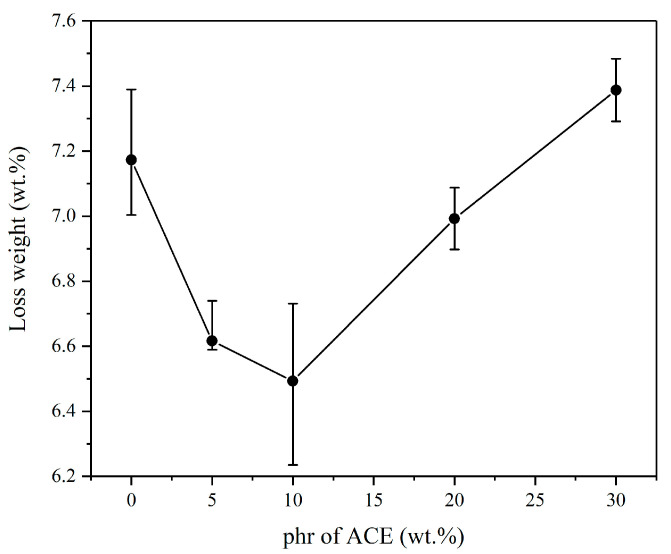
Abrasion testing results of PC/ASA/ACE.

**Figure 7 polymers-14-00782-f007:**
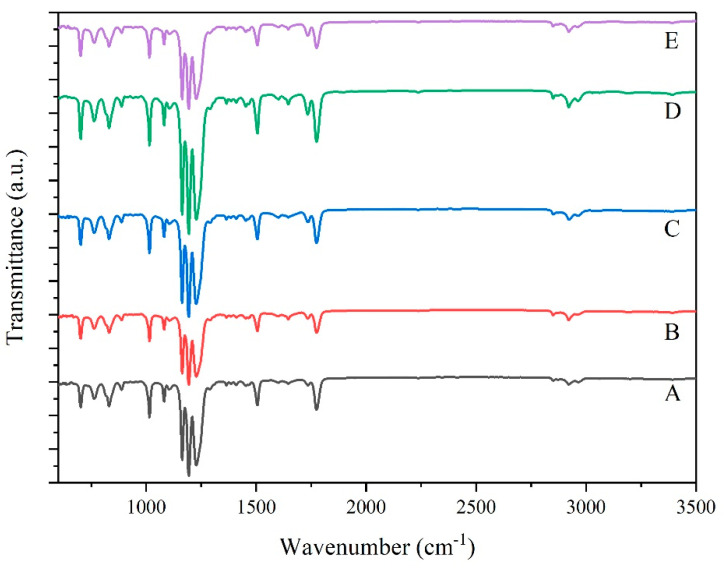
FTIR spectra of PC/ASA/ACE: A: 0 phr, B: 5 phr, C: 10 phr, D: 20 phr, and E: 30 phr.

**Figure 8 polymers-14-00782-f008:**
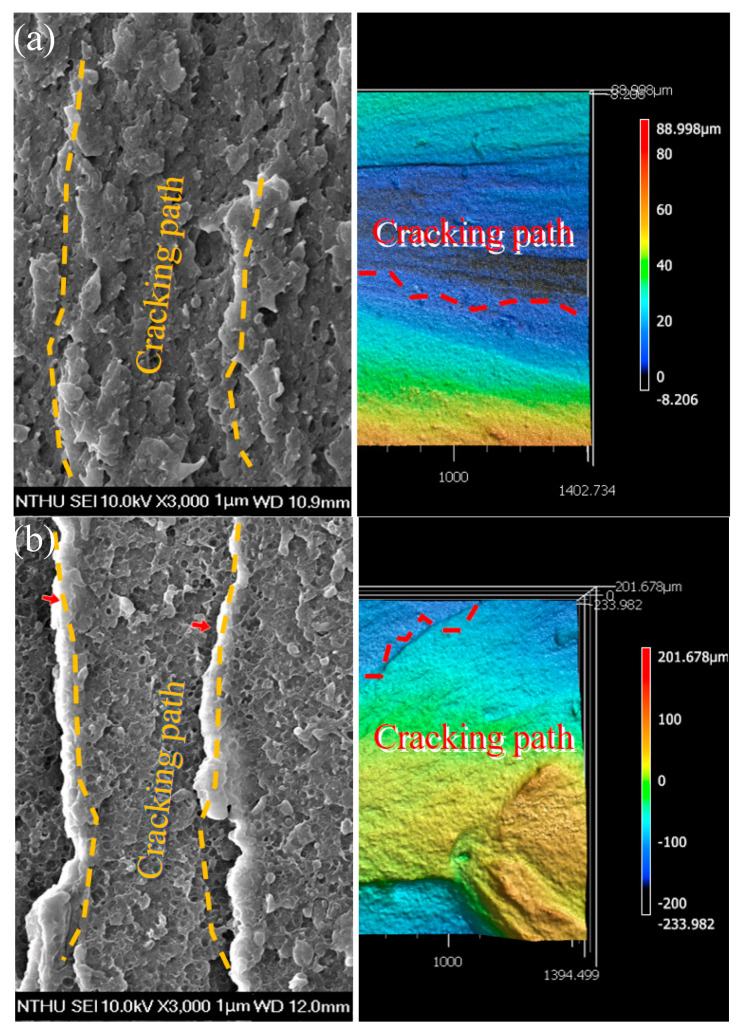
SEM and LSCM images of PC/ASA/ACE: (**a**) 0 phr, (**b**) 5 phr, (**c**) 10 phr, (**d**) 20 phr, and (**e**) 30 phr.

**Figure 9 polymers-14-00782-f009:**
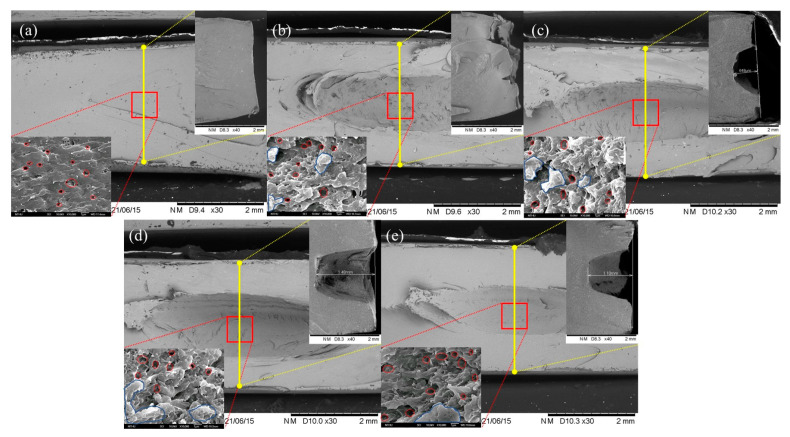
Cross-sectional main view, left view, and 10,000× magnification SEM image of ASA/PC/ACE: (**a**) 0 phr, (**b**) 5 phr, (**c**) 10 phr, (**d**) 20 phr, and (**e**) 30 phr.

**Figure 10 polymers-14-00782-f010:**
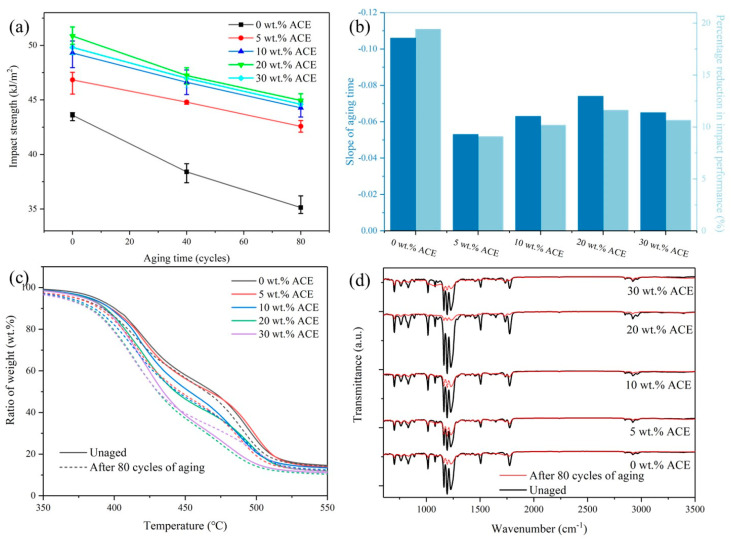
Impact performance of different aging times of PC/ASA/ACE: (**a**) impact strength variation with aging time, (**b**) slope and impact decrease after 80 cycles, (**c**) TG curve variation with different aging times, and (**d**) FTIR with different aging times.

**Table 1 polymers-14-00782-t001:** Experiment design.

Experimental Group	1	2	3	4	5
PC/ASA (wt.%)	70/30	70/30	70/30	70/30	70/30
ACE (wt.%)	0	5	10	20	30

**Table 2 polymers-14-00782-t002:** Temperature parameter table of twin-screw extruder.

Tube 1	Tube 2	Tube 3	Tube 4	Tube 5	Head
245 °C	245 °C	250 °C	255 °C	260 °C	260 °C

**Table 3 polymers-14-00782-t003:** Temperature parameter table of injection molding machine.

Tube 1	Tube 2	Tube 3	Tube 4	Mold
245 °C	250 °C	255 °C	260 °C	70 °C

**Table 4 polymers-14-00782-t004:** Glass transition temperature of PC/ASA/ACE.

Blending System PC/ASA/ACE	Glass Transition Temperature				
	T_g__1_	ΔT_g12_	T_g2_	ΔT_g23_	T_g3_
PC/ASA/ACE(70/30/0)	−38.1	151.4	113.3	37.2	150.5
PC/ASA/ACE(70/30/5)	−33.1	146.3	113.2	33.9	147.1
PC/ASA/ACE(70/30/10)	−34.2	147.6	113.4	34.3	147.7
PC/ASA/ACE(70/30/20)	−35.1	147.8	112.7	34.6	147.3
PC/ASA/ACE(70/30/30)	−33.7	145.5	111.8	32.9	144.7

**Table 5 polymers-14-00782-t005:** 5% weight loss of PC/ASA/ACE.

Sample	0 wt.% ACE	5 wt.% ACE	10 wt.% ACE	20 wt.% ACE	30 wt.% ACE	100 wt.% ACE
5% weight loss/°C	386.8	382.2	382.7	380.8	379.2	315.5

**Table 6 polymers-14-00782-t006:** Toughness of PC/ASA/ACE.

Sample	0 wt.% ACE	5 wt.% ACE	10 wt.% ACE	20 wt.% ACE	30 wt.% ACE
Toughness (J/m^3^)	58.7	72.1	75.8	81.9	79.7

**Table 7 polymers-14-00782-t007:** Roughness parameters of PC/ASA/ACE (70/30/0–30).

Roughness Parameter	Unit	0 wt.% ACE	5 wt.% ACE	10 wt.% ACE	20 wt.% ACE	30 wt.% ACE
Sa	μm	0.89	2.74	4.95	10.96	3.75
Sz	μm	7.34	13.70	26.71	70.15	23.58
Sdr		0.05	0.08	0.24	0.29	0.08

**Table 8 polymers-14-00782-t008:** Carboxyl index of PC/ASA/ACE (70/30/0-30) at different aging times.

Carboxyl Index	0 wt.% ACE	5 wt.% ACE	10 wt.% ACE	20 wt.% ACE	30 wt.% ACE
Unaged	1.01	1.00	1.01	1.02	1.01
After 80 cycles of aging time	1.58	1.39	1.45	1.48	1.40

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
