# Peer review of "Stiffness Enhancement, Anti-Aging, and Self-Forming Holes in Polycarbonate/Acrylonitrile-Styrene-Acrylic by the Core-Shell Structure of Acrylic Resin"

_polymers, 2022, doi:10.3390/polym14040782_

Round 1

Reviewer 1 Report

In this manuscript, Huang Ji, et al. systematically studied the effect of adding acrylic resin (ACE) as modifier into polycarbonate/acrylonitrile-styrene-acrylic(PC/ASA) resin on its mechanical and weathering resistance performances. Combined with the facture surface morphology analysis and mechanical testing, it is demonstrated the introduction of ACE effectively improves the impact and abrasion resistance of the material, due to the self-forming hole phenomenon. Meanwhile, it is also found the improvement of these properties is achieved at the cost of lowering tensile strength, hardness and melt flowing rate. The results from this work show the potential of applying ACE as an effective impact-resistance modifier in PC/ASA resin system to improve its impact performance, aiming at specific application areas, such as airplane windshield. Although the presented results seem reasonable and valuable, there are several concerns prevent the manuscript from being accepted for publication in Polymers at this moment.

  • On page 4 and Fig.1b, the author states the increase of the peak value of tanδ from DMA indicates the enhancement of material toughness, which is not accurate. The increase of tanδ value shows the improved energy dissipating ability of material, while the toughness is material’s resistance to fracture and, as such, is measured as the energy required to cause fracture [1], which should be obtained from static stress-strain curve.
  • I suggest the author to include the stress-strain curves from tensile testing of PC/ASA/ACE materials in the manuscript, which can clearly show the effect of ACE inclusion on the mechanical performances of the resin system. Moreover, the author should calculate the toughness of material from the corresponding stress-strain curves and analyze the effects of adding ACE on material toughness based on the results.
  • The schematic diagrams shown in Fig.5 and Fig.11 are very similar, which should be combined in one figure. Moreover, this diagram is very confusing, the regularly packing of spheres in the polymer blends looks like the crystalline structures formed in metallic materials, which is not an accurate schematic diagram for the amorphous thermoplastic PC/ASA blend resin systems.
  • There are several typos in the manuscript, please carefully proofread the whole manuscript and correct them. Such as, in line 78, “Direct mechanical analysis (DMA)” and in line 116, “N2”.

[1] R.O. Ritchie, The conflicts between strength and toughness, Nat. Mater. 10(11) (2011) 817-822.

Reviewer 2 Report

The subject of the study is interesting and could be of help to those who wish to work in this field. However, the results and discussion section is often limited to simple observations and comments on the characterizations made and lacks scientific analysis. Moreover, it is difficult to link the influence of the composition of the mixture, the morphology, the basic characterizations to the thermo-mechanical properties and to the "self-forming" hole phenomenon.
The overall organization of the paper does not seem to be appropriate for the study, and the experiment-by-experiment description of the results detracts from the overall understanding of the paper. Moreover, the English language needs to be revised, some terms are clumsy, some sentences are not understandable which harms the quality of the proposed study.

The methods of analysis lack precision and detail, the basic materials should be described in more detail, and their properties well specified. Some of the arguments need to be revised, and rewritten to suit what is expected from a scientific publication.

The introduction should be revised and organized in four paragraphs (generalities, state of the art and context of the study, lack in the state of the art, and content of the present article), as the one presented does not put forward the originality and novelty of the study. It focuses too much on general aspects, as underlined by the small number of references cited. (no tables in an introduction)

The "Materials and Methods" section should be completed, justifying the approaches and objectives of each of the methods, as well as the experimental protocol in such a way as to ensure that this study can be reproduced. The use of the melt flow index would be more appropriate in the study.

Concerning the "Results" part, all elements or parts of text dealing only with generalities should be avoided. Rather than dealing with test by test, it would be desirable to approach the writing from another aspect. What is quite surprising is that the basic mixture, and the influence of the ACE rate, on the morphology, phase miscibility, the organization of polymers in relation to each other, interracial energies, ... is not studied beforehand. Moreover, a DSC analysis part is missing, which would eventually allow to highlight the crystallization aspects that are not treated here but only assumed. Most of the figures are blurred and should be improved, and the legend of them should be more explicit. Some employees do not make sense in the context of the study. The document should be corrected and proofread by a native to ensure that it is understandable to the community. 
The data provided, especially temperatures or other values, should be rounded to the unit, taking into account the error of manipulation or the precision of the measurement. 

Round 2

Reviewer 1 Report

In the revised manuscript, concerns and questions raised in the 1st  review for the initial manuscript have been addressed and answered by the authors. The modification to the context makes this work to be more clear and solid. Therefore, I suggest the publication of this paper in Polymers.

Author Response

Thank you very much for your suggestions in the research and writing of this paper.

Reviewer 2 Report

The new version of the paper is more clear and have been significantly improved in the form. Nevertheless, some choices have been made which lead incomplete analysis of the results. The standard deviation for all the measurement should be added
Thus, if there are not DSC results all the comments about the crystallinity should be avoided. the titles of the different parts need to be modified (and probably the organization of the manuscript) to be more representative of the content. It is rather awkward to use as a title the methods of characterization, which does not allow to make a link between all the parts.
Regarding part 3.2. Melt flow rate.
The term viscosity should be changed to fluidity. Loss modulus was missing.
The tests were conducted at 150°C which is a temperature very close to Tg3. On the Tan delta curves, we can see that the properties of the mixtures are not the same at this temperature, and that the relaxations of the macromolecular chains are still present. This can explain the variations of the MFR. Rheological explanations on the behavior of macromolecular chains must be added.

Part 3.3 - Thermal stability
This is not a "glass transition diagram".
For a more complete analysis it would be necessary to have the TGA curve of the ACE, as well as the derived curves. The curves of mass differences versus temperature could also be added to analyze

Part 3.8 -
it would have been desirable to do an FTIR and TG analysis to determine the chemical modifications during the "Accelerated weathering" tests. It also lacks explanations on the evolution of the macromolecular chain behavior.

An analysis of the distribution of ACE within the material is missing, which does not allow to highlight their effects on "stress concentration
